# Metabolic Fate of Orally Ingested Proanthocyanidins through the Digestive Tract

**DOI:** 10.3390/antiox12010017

**Published:** 2022-12-22

**Authors:** Yoshimi Niwano, Hidetsugu Kohzaki, Midori Shirato, Shunichi Shishido, Keisuke Nakamura

**Affiliations:** 1Faculty of Nursing, Shumei University, 1-1 Daigaku-cho, Yachiyo, Chiba 276-0003, Japan; 2Department of Advanced Free Radical Science, Tohoku University Graduate School of Dentistry, 4-1 Seiryo-machi, Aoba-ku, Sendai 980-8575, Japan

**Keywords:** proanthocyanidin, metabolic fate, digestive tract, gut microbiota

## Abstract

Proanthocyanidins (PACs), which are oligomers or polymers of flavan-3ols with potent antioxidative activity, are well known to exert a variety of beneficial health effects. Nonetheless, their bioaccessibility and bioavailability have been poorly assessed. In this review, we focused on the metabolic fate of PACs through the digestive tract. When oligomeric and polymeric PACs are orally ingested, a large portion of the PACs reach the colon, where a small portion is subjected to microbial degradation to phenolic acids and valerolactones, despite the possibility that slight depolymerization of PACs occurs in the stomach and small intestine. Valerolactones, as microbiota-generated catabolites of PACs, may contribute to some of the health benefits of orally ingested PACs. The remaining portion interacts with gut microbiota, resulting in improved microbial diversity and, thereby, contributing to improved health. For instance, an increased amount of beneficial gut bacteria (e.g., *Akkermansia muciniphila* and butyrate-producing bacteria) could ameliorate host metabolic functions, and a lowered ratio of Firmicutes/Bacteroidetes at the phylum level could mitigate obesity-related metabolic disorders.

## 1. Introduction

Proanthocyanidins (PACs), also known as condensed tannins, are substances that produce red anthocyanidin pigments when decomposed by acid and are oligomers or polymers of flavan-3-ols, such as epicatechin and catechin. They are widely distributed in fruits, grains, and leaves [1,2,3,4,5], especially in cocoa, black soybeans, cinnamon, apples, and grape seeds [6]. In addition, grape seed PACs have an average degree of polymerization (DP) between 2 and 17 [1].

We previously reported that grape seed PACs have direct antioxidant potential in vitro against di(phenyl)-(2,4,6-trinitrophenyl)iminoazanium (DPPH; a stable radical), superoxide anion radical (O_2_^−•^), hydroxyl radical (^•^OH), singlet oxygen (^1^O_2_), and hydrogen peroxide (H_2_O_2_) [7]. In oxidative-stress-induced cells, PACs significantly improved antioxidant enzyme activities (e.g., glutathione peroxidase, superoxide dismutase, and catalase), leading to decreased levels of reactive oxygen species and malondialdehyde [8]. In addition, they significantly activate the nuclear factor–erythroid 2-related factor 2 (Nrf2)/antioxidant response element (ARE) pathway, including the increased expression of NAD(P)H:quinone acceptor oxidoreductase 1 and heme oxygenase 1. These characteristic features observed in vitro are thought to contribute to various therapeutic effects, including anti-adipogenesis in adipocytes [9], anti-cancer effects in several cancer cells [10,11,12,13,14], and neuroprotective effects in rat pheochromocytoma cells (PC12 cells) [15,16,17,18].

In in vivo studies, PACs alleviated severe acute pancreatitis in mice via their anti-inflammatory properties [19], exerted anti-obesity and anti-diabetic activity in a type 2 diabetes model of KKAy mice [20] and anti-obesity activity in a mouse model of high-fat-diet-induced obesity [21], and showed neuroprotective activity in zebrafish and rat models of Parkinson’s disease [8,22]. We previously demonstrated that orally administered grape seed PACs prevented bone loss in the lumbar vertebrae and femur in ovariectomized (OVX) mice, and they ameliorated the healing of defects created on the calvaria and osseointegration of a tibial implant in OVX rats, likely by counteracting the accelerated osteoclastogenic activity induced by estrogen deficiency [23]. To attain a better understanding of such health-beneficial activities, pharmacokinetic analysis is imperative. However, there is a paucity of evidence related to the structural complexity of PACs. Thus, in this review, we focused on the bioaccessibility and bioavailability of PACs by exploring their metabolic fate through the digestive tract.

## 2. Basic Structures of Proanthocyanidins (PACs)

Flavan-3-ols have a basic structure consisting of A, B, and C rings, in which 3, 5, 7, 3′, or 4′ is hydroxylated. For example, the 3-hydroxylated group has two conformations: the 2,3-cis isomer is (−)-epicatechin, and the 2,3-trans isomer is (+)-catechin. Oligomers are formed by C4-C8 or C2-O-C7 bonds between monomers with these basic structures. The isomers are roughly divided into two groups according to their binding modes—those with C4-C8 or C4-C6 bonds are called B-type, and those with additional C2-O-C7 bonds are called A-type (Figure 1). Naturally occurring B-type PACs are predominant in plants such as cocoa, bayberry, and grapes [24,25,26]. Concerning PAC dimers, the A-type dimers and B-type dimers are numbered as A1, A2, B1, and B2; e.g., B1 consists of (−)-epicatechin (C4-C8) (+)-catechin, and B2 consists of (−)-epicatechin (C4-C8) (−)-epicatechin. Apart from the A- and B-type dimers, PAC C1 (epicatechin-(C4-C8)-epicatechin-(C4-C8)-epicatechin) and PAC C2 (catechin-(C4-C8)-catechin-(C4-C8)-catechin) are trimeric and belong to the group of B-type PACs. PACs are also divided into three categories: procyanidins (oligomeric PACs formed from catechin and epicatechin), propelargonidins (from afzelechin and epiafzelechin), and prodelphinidins (from gallocatechin and epigallocatechin) [27]. Based on the DP, PACs with a low DP are called oligomers, and those with a high DP are called polymers. For instance, previous papers defined oligomers and polymers as structures with DP values of four to ten and those with more than ten, respectively [28,29].

Regarding the DP and stereochemistry of oligomeric PACs, the condensation of monomeric flavan-3-ol units compactly forms a helical PAC structure in an aqueous solution [30,31,32], leading to interactions between saliva proteins, which causes astringency in wine-tasting processes [33]. In addition, the hydrophobicity of PACs, as measured with the octanol–water partition coefficient (logP), significantly decreases with an increase in the degree of polymerization due to the large number of phenolic hydroxyl groups covering them [34,35]. In more detail, PACs with a higher DP have a helical structure comprising a hydrophilic surface covered by a large number of hydroxyl groups; their internal region is hydrophobic, making them likely to interact with biogenic substances, such as proteins, peptides, carbohydrates, lipids, and oligonucleotides [36].

## 3. Pharmacokinetics of Proanthocyanidins (PACs)

### 3.1. Oral Stability

Interactions between PACs and biogenic substances in the oral cavity vary. As is the case with biogenic substance–phenolic compound interactions [37,38,39], PACs can interact with carbohydrate polymers via hydrogen bonding, leading to the formation of non-digestible amylose–PAC complexes [40,41,42]. In a previous report, sorghum PACs were extractable after cooking with starches that varied in amylose content [43]. If PACs and carbohydrate polymers interact hydrophobically and/or through hydrogen bonds, the PACs are likely to be extractable. PACs also inhibit α-amylase due to a non-covalent hydrophobic interaction with the enzyme [42,44,45]. Thus, when PACs are orally taken, their bioavailability could be affected depending on the intradigestive environment. In an in vitro oral digestion study where 5 mL of simulated saliva fluid composed of amylase enzyme was applied to Chinese bayberry leaf PACs, the PAC dimers showed no significant differences during in vitro digestion, whereas the trimers were significantly decreased after 2 min of oral digestion [46]. Concomitantly, the flavan-3-ol monomers probably increased due to the degradation of the trimers. However, salivary proteins (proline-rich proteins and histatins) are known to have an affinity to PACs [47,48,49], irrespective of the amylase–PAC interactions, with the salivary protein–PAC complexes being present in the stomach. The protein–PAC complexes that deposit in the stomach then separate due to the acidic environment; for example, PAC trimer–amylase complexes were reported to separate in the gastric environment of the stomach, resulting in an increase in trimer content [46].

### 3.2. Gastric Stability

To investigate the gastric stability of PACs, several in vitro studies using simulated gastric juice were conducted, but the results were controversial. PAC oligomers (trimer to hexamer) purified from cocoa were hydrolyzed to mixtures of epicatechin monomer and dimer [50], apple dimeric PAC B2 was almost completely degraded into (−)-epicatechin [51], and the PAC content in an extract of *Hypericum perfoliatum* L. significantly decreased by 25% [52]. On the contrary, other studies reported that PACs with a high DP (mean DP ≥ 6) from grape seeds were remarkably stable in the gastric environment and did not degrade into more readily absorbable monomers [53,54], PACs from *Acacia mearnsii* remained stable during gastric digestion in vitro [55], and the mean DP of PACs isolated from *Choerospondias axillaris* peel was not affected [56]. A human in vivo study showed that cocoa beverage PACs were stable during gastric transit, with the pH of gastric contents increasing from 1.9 ± 0.2 to 5.4 ± 0.2 after consumption [57]. Regarding the effects of macronutrients, it was reported that a higher fat content or the presence of carbohydrate-rich food did not greatly affect the in vitro gastric stability of PACs [54,58]. In summary, the gastric stability of PACs depends on their types and on the electrolytes used, the dietary source, the duration of exposure to the gastric environment, and the pH conditions of gastric juice [45,55,57]. The timing of oral intake can be an important factor when considering gastric stability. For instance, in the postprandial state, PACs were present with a mixture of foodstuffs and gastric juice under acidic conditions. However, in the fasting state, there was little gastric juice with slightly higher pH conditions because the acid secretion (a pH of 2 under basal conditions with an empty stomach) was buffered by the food bolus [57]. Collectively, PACs are depolymerized to some extent under gastric conditions and then pass into the small intestine.

### 3.3. Small-Intestinal Stability and Absorption

The first step after gastric digestion is exposure to pancreatic juice in the duodenum. It was reported that slight depolymerization of PACs could be observed in an in vitro small-intestinal model that used pancreatic enzymes and bile salts [54]. A similar in vitro study showed that the mean DP of PACs was slightly decreased, which was possibly due to interactions with digestive enzymes [56]. Collectively, PACs were rather chemically stable with respect to depolymerization during their passage through the simulated duodenal digestion. Regarding intestinal absorption of PACs, it was reported that the A1, A2, and B2 PAC dimers were slightly absorbed without conjugation or methylation from the small intestine in an in situ perfusion model of the rat small intestine [59]. Similarly, a study on the absorption rate of PACs without digestion, which was measured with the Caco-2 monolayer transport assay, showed that PAC dimers could traverse the Caco-2 monolayer [46], and trimers and tetramers could be transported across Caco-2 cells at low rates [60].

### 3.4. Colonic Stability and Absorption

When PACs reach the colon, they are likely to be affected by gut microbiota. In in vitro fermentation of grape seed extracts that were rich in B-type PACs, the maximum formation of intermediate metabolites, such as valerolactones, valeric acid, several phenolic acids, and gallic acid, was observed at 5–10 h of incubation with fecal microbiota. Subsequently, the incubations (10–48 h) resulted in the appearance of mono- and non-hydroxylated forms of previous metabolites, which was likely due to dehydroxylation reactions [61,62]. These in vitro results were also consistent with those from a human study. When humans consumed a test drink containing PACs with a DP ranging from 2 to 10, γ-valerolactones were mainly detected in the plasma [63], thus rejecting the notion that PACs are broken down into flavanols prior to their absorption. In another human study that was conducted to comparatively investigate the metabolic fate of (−)-epicatechin, PAC B1 (a dimer) and polymeric PACs, all of which were encapsulated in hard gelatin to minimize interactions with the oral and gastric environments, it was observed that free PAC B1, 4-hydroxy-5-(3′,4′-dihydroxyphenyl)valeric acid, 5-(3′,4′-dihydroxyphenyl)-valerolactone were detected in the plasma after PAC B1 ingestion, but no dimeric or oligomeric PACs were detected in the plasma after the ingestion of polymeric PACs with a mean degree of polymerization of 5.9 [64]. Thus, 5-(3′,4′-dihydroxyphenyl)-valerolactone is a dominant in vivo metabolite of PAC B1 produced by the gut microbiota. Moreover, small portions of PAC B1 were metabolized by the phase II metabolism after entering into circulation. In addition, microbial degradation would be hampered because of the low uptake of compounds by bacteria due to their huge molecular size. These findings were consistent with those of two rat studies that showed that ingested polymeric PACs were present in the colon as the intact parent compounds, and they were responsible for the local beneficial biological actions [65,66].

Phenyl-γ-valerolactones, as microbiota-generated catabolites of PACs, have been shown in preclinical studies to have some potential health-beneficial effects, such as reducing the risk of colorectal cancer [67] and neuroprotection by regulating intracellular proteolysis [68]. In a study in which the cellular antioxidant effect of polyphenol metabolites was examined, 5-(3′,4′-dihydroxyphenyl)-γ-valerolactone showed the highest antioxidant effect among the investigated polyphenol metabolites [69]. It was also shown that 5-(3′,4′-dihydroxyphenyl)-γ-valerolactone had catalase-like activity and promoted the Nrf2/ARE pathway. Collectively, γ-valerolactones produced from PACs by gut microbiota may contribute to some health-beneficial effects following oral ingestion of PACs.

### 3.5. Effects on Gut Microbiota

Apart from bacterial transformation, PACs could affect the gut microbiota. Although most in vivo studies were conducted to investigate the effects of PACs on altered gut microbiota under certain pathological conditions, a few studies using normal animals have been conducted. It was reported that dietary PACs for 6 days resulted in an ecological shift in the microbiome, dramatically increasing the operational taxonomic units (OTUs) of *Lachnospiraceae,* unclassified *Clostridales, Lactobacillus,* and *Ruminococcus* in crossbred female pigs. Further, intact parent PACs (dimer-pentamer) and major phenolic metabolites (4-hydroxyphenylvaleric acid and 3-hydroxybenzoic acid) were found in feces [70]. It was reported that *Lachnospiraceae* and *Ruminococcus* are the major butyrate and propionate producers in human fecal samples [71], and butyrate can modulate oxidative stress in the colonic mucosa of healthy humans [72]. In a review article, butyrate was reported to lead to more specific and efficacious therapeutic strategies for the prevention and treatment of different diseases ranging from genetic/metabolic conditions to neurological degenerative disorders [73]. In particular, in a human study, the transfer of intestinal microbiota from lean donors increased insulin sensitivity in individuals with metabolic syndrome along with levels of butyrate-producing intestinal microbiota, suggesting that intestinal microbiota should be developed as therapeutic agents for increasing insulin sensitivity in humans [74]. If PACs have the ability to increase butyrate producers, they may work not only for colonic health, but also for systemic health. Another study using weaned piglets revealed that dietary grape seed PACs improved the microbial diversity in ileal and colonic digesta, with the most abundant OTUs belonging to two phyla: Firmicutes and Bacteroidetes [75]. The PACs also decreased the abundance of *Lactobacillaceae* and increased that of *Clostridiaceae*, accompanied by improved intestinal mucosal barrier function and increased concentration of propionic and butyric acids in the intestinal digesta. In a rat study in which an 8-day oral gavage of grape seed PACs (monomeric (21.3%), dimeric (17.4%), trimeric (16.3%), tetrameric (13.3%), and oligomeric (31.7%)) was administered to normal female rats, the ratio of Firmicutes to Bacteroidetes at the phylum level was lowered with increased plasma glucagon-like-peptide-1 level [76]. More recently, it was reviewed that PACs have a prebiotic and antimicrobial role that favors homeostasis of the intestinal environment, thus reducing the survival of Gram-negative bacteria that produce lipopolysaccharide (LPS) [77]. As LPS triggers the activation of the Toll-like receptor-4 (TLR-4) inflammatory pathway, PACs can minimize endotoxemia.

As for animal studies under pathological conditions, most studies applied high-fat diet (HFD)- or high-fat/high-sucrose diet (HFHSD)-induced metabolic syndrome model animals. PAC-rich grape seed/pomace extract [78,79,80], PAC-rich cranberry extract [81], and apple PACs [82] showed improved symptoms of metabolic syndrome concomitantly with an altered gut microbial environment. Some studies revealed that PACs increase *Akkermansia muciniphila* [78,80] or *Akkermansia* at the genus level [82], the former of which is a well-known beneficial gut bacterium that improves host metabolic functions and immune responses [83,84,85,86,87,88,89]. Accounting for 3–5% of the microbial community in healthy individuals, *A. muciniphila* is a mucinolytic bacterium found in the mucus layer of the human gut [90], and it has the potential to restore mucus thickness and intestinal barrier integrity [91,92]. This bacterium also has the ability to decrease the progression of many diseases, such as obesity and type 2 diabetes mellitus [93,94]. As such, *A. muciniphila* is considered a promising probiotic candidate [88]. At the phylum level, PACs could decrease the ratio of Firmicutes/Bacteroidetes [79,82]. The dominant gut microbial phyla are Firmicutes, Bacteroidetes, Actinobacteria, Proteobacteria, Fusobacteria, and Verrucomicrobia, with the first two phyla being the most common in healthy human individuals [95]. Phylum-level analyses of Firmicutes and Bacteroidetes have shown that they are associated with obesity and that an increased population of Bacteroidetes, as well as a reduced population of Firmicutes, could improve obesity [96,97,98,99,100]; this is likely via the depression of the increased capacity for energy harvesting from the diet [99]. In a human study, the relative proportion of Bacteroidetes was decreased in obese people in comparison with that in lean people, and this proportion increased with weight loss with two types of low-calorie diets [101]. These findings indicate that obesity is associated with a microbial component, paving the way for investigations into the potential therapeutic implications of gut microbiota. Aside from HFD- or HFHSD-fed animals, PACs normalized the imbalanced Firmicutes/Bacteroidetes ratio observed in OVX mice in a menopause model and prevented OVX animals from having an increased weight [102].

If the microbial degradation of PACs is hampered because of the low compound uptake by bacteria due to their huge molecular size, how exactly they affect the gut microbiota becomes the primary concern. Some PACs exert antimicrobial activities by preventing bacterial adhesion to human cells [103,104], with PAC-rich cranberry being used clinically as an adjuvant therapy in the prevention of urinary tract infections [105]. It has also been reported that anti-adhesion activity could be challenging in the development of new antimicrobials that are able to withstand the increasing repertoire of bacterial resistance [106]. In dentistry, PACs are known to have specific antibacterial characteristics of attacking periodontopathogenic bacteria (*Porphyromonas gingivalis*), but not oral commensal bacteria (*Streptococcus salivarius*) [107,108]. PACs’ antibacterial activity in the oral cavity may be attributed to their biofilm-disrupting properties by interfering with the N-acylhomoserine lactone-mediated quorum sensing of the bacteria, which tightly regulates the expression of multiple virulence factors in opportunistic pathogenic Gram-negative bacteria [109,110]. Thus, PACs could be some of the substances affecting gut microbiota via antibacterial activity. Further studies are needed to clarify the effects of PACs on the gut microbiota.

The aforementioned metabolic fate of PACs through the digestive tract and their health-beneficial effects in association with gut microbiota are summarized in Figure 2 and Figure 3, respectively.

## 4. Conclusions

A large portion of orally ingested oligomeric and polymeric PACs reach the colon, where a small portion of them are subjected to microbial degradation into phenolic acids and valerolactones. The rest interact with gut microbiota, resulting in improved microbial diversity, which includes an increased amount of beneficial gut bacteria (e.g., *Akkermansia muciniphila*), which could ameliorate host metabolic functions, and a lowered ratio of Firmicutes/Bacteroidetes at the phylum level, which could mitigate obesity-related metabolic disorders. Further, PACs have the potential to increase butyrate-producing microbiota and decrease LPS-producing bacteria, leading to the prevention and treatment of different diseases ranging from metabolic conditions to neurological degenerative disorders. These microbial changes could confer some of PACs’ health-beneficial effects.

## Figures and Tables

**Figure 1 antioxidants-12-00017-f001:**
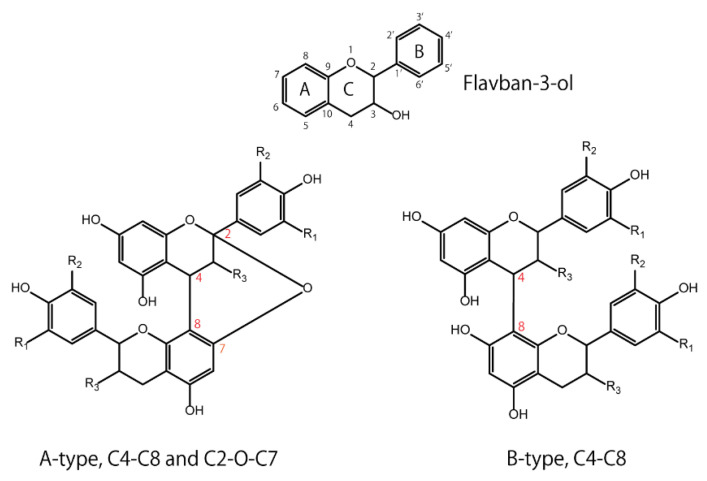
Basic structures of flavan-3-ol: A-type and B-type proanthocyanidins.

**Figure 2 antioxidants-12-00017-f002:**
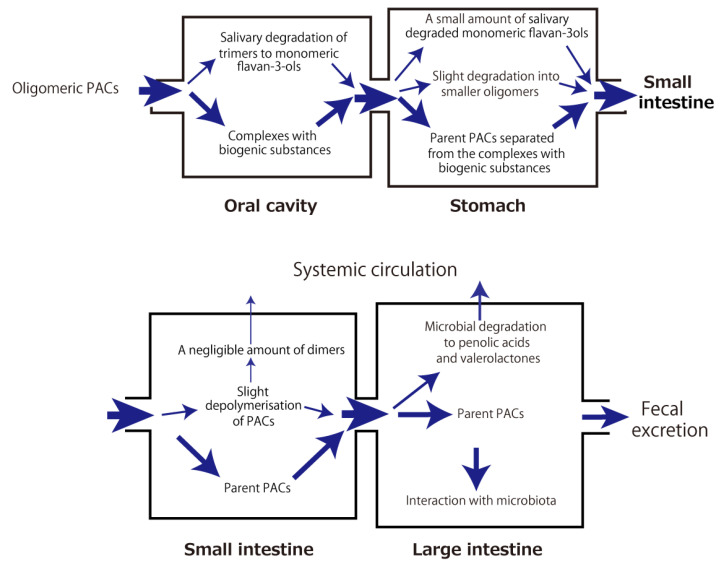
Intradigestive fate of orally ingested proanthocyanidins (PACs).

**Figure 3 antioxidants-12-00017-f003:**
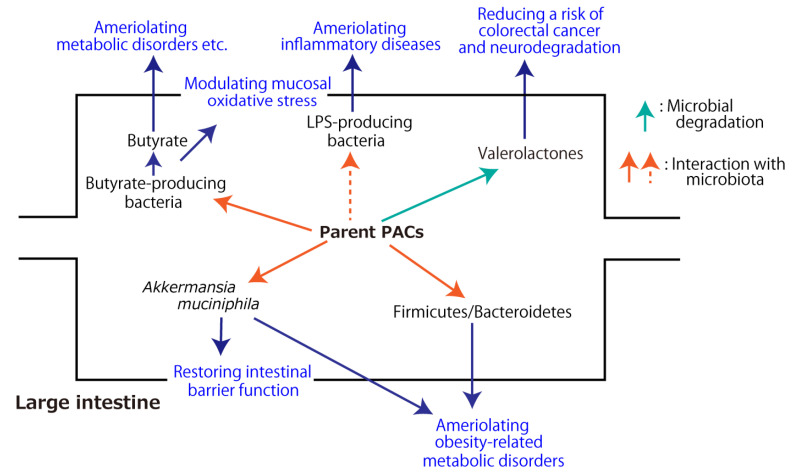
Health-beneficial effects of proanthocyanidins (PACs) associated with gut microbiota. The dashed arrow indicates the inhibitory effect.

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
