# Peer review of "Metabolic Fate of Orally Ingested Proanthocyanidins through the Digestive Tract"

_antioxidants, 2022, doi:10.3390/antiox12010017_

Round 1

Reviewer 1 Report

The manuscript does not reach the required quality standard of this journal. I am only going to list a few motives for my decision.

In my opinion the introduction is very short and could be used to highlight the health benefits of PACs. I would advise the authors to be careful in the use of connectors (besides, moreover, in adition) I have the feeling these are used erratically.  Please work on the flow of the text, connecting the ideas is important.

*In section 2, Basic structures of proanthocyanidins (PACs), the way the structures of PACs are described makes it difficult to understand and I feel is incomplete. They mention the classification based linkage and the stereoconfiguration between monomers type A and type B. The authors do not mention that PAsC can be divided into three categories according to differences in subunit composition: procyanidin with the subunit catechin or epicatechin, propelargonidins with afzelechin, and prodelphinidins with gallocatechin or epigallocatechin. Also there are different PACs based on the degree of polymerization; known as oligomers if they have a low degree of polymerization and polymers with a high degree of polymerization. This is important since the absorption of these compounds is highly dependent on the polymerization degree among other things such as the food matrix. I understand the information is there (L62-L71) but it is difficult to understand the way it is presented.

*The information in section 3.1 Oral stability is very confusing.

*In section 3.2 Gastric stability. Once again I find all the information unclear, needs to be worked and discussed.

L99 high degree of polymerization >2 should be >6.

How can reference 42 (Am J Clin Nutr 2002) in vivo studies support the in vitro studies of references 39-41 done later (Food Chem 2013, Br J Nutr 2010, Food Res Int 2015).

*In the section 3.5 Effects on gut microbiota I feel there is a lack of information. I am sure there are more recent papers published on the positive effect of PACs on gut microbiota.

In my opinion when referring to healthy animal models it is not correct to say normal animals. Line 155 change normal animals for healthy animals or simply remove this I don’t think it is necessary to include subsections.

Author Response

The manuscript does not reach the required quality standard of this journal. I am only going to list a few motives for my decision.

Concern of the reviewer

In my opinion the introduction is very short and could be used to highlight the health benefits of PACs. I would advise the authors to be careful in the use of connectors (besides, moreover, in addition) I have the feeling these are used erratically.  Please work on the flow of the text, connecting the ideas is important.

Response to the comment

We would like to thank you for your comment. As you kindly pointed out, lack of enough health benefits of PACs makes the introduction poor. So, we highlighted much more to the health benefit effects. Also, we checked the frequent use of connections that makes the content of paper rather redundant.

Revised manuscript, line 24-54

  1. Introduction

Proanthocyanidins (PACs), also known as condensed tannins, are substances that pro-duce red anthocyanidin pigments when decomposed by acid and are oligomers or pol-ymers of flavan-3-ols such as epicatechin and catechin. They are widely distributed in fruits, grains, and leaves [1-5], especially in cocoa, black soybeans, cinnamon, apples, and grape seeds [6]. Besides, the grape seed PACs have an average degree of polymeri-zation (DP) between 2 and 17 [1].

We previously reported that grape seed PACs have direct antioxidant potential in vitro against di(phenyl)-(2,4,6-trinitrophenyl)iminoazanium (DPPH; a stable radical), superoxide anion radical (O2 –•),  hydroxyl radical (OH), singlet oxygen (1O2), and hy-drogen peroxide (H2O2) [7]. In oxidative stress-induced cells, PACs significantly improved antioxidant enzyme activities (e.g., glutathione peroxidase, superoxide dis-mutase, and catalase), leading to decreased levels of reactive oxygen species and malondialdehyde [8]. In addition, they significantly activated the nuclear factor-erythroid 2-related factor 2 (Nrf2)/antioxidant response element (ARE) pathway such as increased the expression of NAD(P)H:quinone acceptor oxidoreductase 1 and heme oxygenase 1. These characteristic features observed in in vitro are thought to contribute to various therapeutic effects, including anti-adipogenesis in adipocytes [9], anti-cancer effects in several cancer cells [10-14], and neuroprotective effects in rat pheo-chromocytoma cells (PC12 cells) [15-18].

In in vivo studies, PACs alleviated severe acute pancreatitis in mice via their anti-inflammatory properties [19], exerted anti‑obesity and anti‑diabetic activity in type 2 diabetes model of KKAy mice [20], anti-obesity activity in a mouse model of high fat diet induced obesity [21], and showed neuroprotective activity in zebrafish and rat models of Parkinson's disease [8, 22]. We previously demonstrated that orally administered grape seed PACs prevented bone loss in the lumbar vertebrae and femur in ovar-iectomized (OVX) mice, and ameliorated healing of defects created on the calvaria and osseointegration of a tibial implant in OVX rats, likely through counteracting the accel-erated osteoclastogenic activity induced by the estrogen deficiency [23]. To get better understanding of such health-benefit activities, pharmacokinetic analysis is imperative. However, there is a paucity of evidence related to the structural complexity of PACs. Thus, through this review, we focused on the bioaccessibility and bioavailability of PACs by exploring their metabolic fate through the digestive tract.

Concern of the reviewer

  1. Basic structures of proanthocyanidins (PACs)

*In section 2, Basic structures of proanthocyanidins (PACs), the way the structures of PACs are described makes it difficult to understand and I feel is incomplete. They mention the classification based linkage and the stereoconfiguration between monomers type A and type B. The authors do not mention that PAsC can be divided into three categories according to differences in subunit composition: procyanidin with the subunit catechin or epicatechin, propelargonidins with afzelechin, and prodelphinidins with gallocatechin or epigallocatechin. Also there are different PACs based on the degree of polymerization; known as oligomers if they have a low degree of polymerization and polymers with a high degree of polymerization. This is important since the absorption of these compounds is highly dependent on the polymerization degree among other things such as the food matrix. I understand the information is there (L62-L71) but it is difficult to understand the way it is presented.

Response to the comment

We appreciate your helpful comment very much. According to your comment, we added some information on PAC C1 and C2, procyanidins, propelargonidins, and prodelphinidins, and oligomers and polymers. As for line 62-71, we rephrased the corresponding part.

Revised manuscript, line 64-74

Concerning PAC dimers, the A-type dimers and B-type dimers are numbered as A1, A2, B1, B2, e.g., B1 is consist of (-)-epicatechin (C4-C8) (+)-catechin, and B2 is (-)-epicatechin (C4-C8) (-)-epicatechin. Apart from the A- and B-type dimers, PAC C1 [epicatechin-(C4-C8)-epicatechin-(C4-C8)-epicatechin] and PAC C2 [catechin-(C4-C8)-catechin-(C4-C8)-catechin] are trimeric and belong to the B-type PACs. PACs are also divided into three categories, viz. procyanidins (oligomeric PACs formed from cate-chin and epicatechin), propelargonidins (from afzelechin and epiafzelechin), and prodelphinidins (from gallocatechin and epigallocatechin) [27]. Based on the DP, PACs with a low DP are called oligomers, and those with a high DP called polymers. For in-stance, previous papers defined oligomers and polymers with DP of four to ten and more than ten, respectively [28, 29].

Revised manuscript, line 94-103

Regarding the DP and stereochemistry of oligomeric PACs, condensation of monomeric flavan-3-ol units compactly forms a helical PAC structure in an aqueous solution  [30-32], leading to interactions between saliva proteins causing astringency in wine-tasting processes [33]. Besides, the hydrophobicity of PACs, as measured by octanol-water partition coefficient (logP) significantly decreases with an increase in its degree of polymerization due to a large number of phenolic hydroxyl groups covering them [34, 35]. More in detail, PACs with a higher DP have a helical structure comprising a hydrophilic surface covered by a large number of hydroxyl groups; their internal region is hydrophobic, making them likely interact with biogenic substances such as proteins, peptides, carbohydrates, lipids, and oligonucleotides [36].

Concern of the reviewer

*The information in section 3.1 Oral stability is very confusing.

Response to the comment

Thank you for your comment. We amended the relevant part a lot to improve readability.

Revised manuscript, line 106-125

3.1 Oral stability

Interactions between PACs and biogenic substances in the oral cavity vary. As is the case with biogenic substance–phenolic compound interactions [37-39], PACs can interact with carbohydrate polymers via hydrogen bonding leading to the formation of non-digestible amylose-PAC complexes [40-42]. In a previous report, sorghum PACs were extractable after cooking with starches varying in amylose content [43], if PACs and carbohydrate polymers interact hydrophobically and/or through hydrogen bonds, the PACs likely are extractable. PACs also inhibit α-amylase due to a non-covalent hydrophobic interaction with the enzyme [42, 44, 45]. Thus, when PACs are orally taken, their bioavailability could be affected depending on the intradigestive environment. In an in vitro oral digestion study where 5 mL of simulated saliva fluid composed of amylase enzyme was applied to Chinese bayberry leaf PACs, PAC dimers showed no significant differences during in vitro digestion, whereas trimers were significantly de-creased after 2 min of oral digestion [46]. Concomitantly, flavan-3-ol monomers probably increased due to the degradation of trimers. However, salivary proteins (proline-rich proteins and histatins) are known to have an affinity to PACs [47-49], irrespective of the amylase-PACs interaction, with the salivary protein-PAC complexes being pre-sent in the stomach. The protein-PAC complexes that deposit in the stomach then separate due to the acidic environment; for example, PAC trimer-amylase complexes were reported to separate in the gastric environment of the stomach, resulting in an increase in trimer content [46].

Concern of the reviewer

*In section 3.2 Gastric stability. Once again I find all the information unclear, needs to be worked and discussed.

Response to the comment

Thank you for your comment. We also amended the relevant part a lot to improve readability.

Revised manuscript, line 127-149

3.2 Gastric stability

To investigate gastric stability of PACs, several in vitro studies using simulated gastric juice were conducted, but the results were controversial. PAC oligomers (trimer to hexamer) purified from cocoa were hydrolyzed to mixtures of epicatechin monomer and dimer [50], apple dimeric PAC B2 was almost completely degraded into (-)-epicatechin [51], and the PACs content in an extract of Hypericum perfoliatum L. significantly decreased by 25% [52]. By contrast, studies reported that PACs with a high DP (mean DP≧6) from grape seeds were remarkably stable in the gastric environment and did not degrade into more readily absorbable monomers [53, 54], PACs from Acacia mearnsii remained stable during in vitro gastric digestion [55], and mean DP of PACs isolated from Choerospondias axillaris peel was not affected [56]. A human in vivo study showed that cocoa beverage PACs were stable during gastric transit, with the pH of gastric contents increasing from 1.9 ± 0.2 to 5.4 ± 0.2 after consumption [57]. Regarding the effects of macronutrients, it was reported that a higher fat content or the presence of carbohydrate-rich food did not greatly affect in vitro gastric stability of PACs [54, 58]. In summary, the gastric stability of PACs depends on their types and electrolytes used, the dietary source, the duration of exposure to the gastric environment, and the pH conditions of gastric juice [45, 55, 57]. The timing of oral intake can be an important fac-tor when considering gastric stability. For instance, in the postprandial state, PACs are present with a mixture of foodstuff and gastric juice under acid conditions. Whereas in the fasting state, there is little gastric juice with slightly higher pH conditions because the acid secretion (pH 2 under basal conditions with an empty stomach) is buffered by the food bolus [57]. Collectively, PACs are depolymerized to some extent under gastric conditions and then pass into the small intestine.

Concern of the reviewer

L99 high degree of polymerization >2 should be >6.

Response to the comment

Thank you very much for your careful reading. We corrected it.

Concern of the reviewer

How can reference 42 (Am J Clin Nutr 2002) in vivo studies support the in vitro studies of references 39-41 done later (Food Chem 2013, Br J Nutr 2010, Food Res Int 2015).

Response to the comment

Thank you very much for your careful reading. As you pointed out, it seems a bit funny. So, we rephrased the relevant part.

Revised manuscript, line 133-139

By contrast, studies reported that PACs with a high DP (mean DP≧6) from grape seeds were remarkably stable in the gastric environment and did not degrade into more readily absorbable monomers [53, 54], PACs from Acacia mearnsii remained stable during in vitro gastric digestion [55], and mean DP of PACs isolated from Choerospondias axillaris peel was not affected [56]. A human in vivo study showed that cocoa beverage PACs were stable during gastric transit, with the pH of gastric contents increasing from 1.9 ± 0.2 to 5.4 ± 0.2 after consumption [57].

Concern of the reviewer

*In the section 3.5 Effects on gut microbiota I feel there is a lack of information. I am sure there are more recent papers published on the positive effect of PACs on gut microbiota.

Response to the comment

We appreciate your constructive comment. We amended the relevant part a lot to improve readability. We also added Figure 3 as make the whole part be better for understanding.

Revised manuscript, line 199-274

Apart from bacterial transformation, PACs could affect the gut microbiota. Alt-hough most in vivo studies were conducted to investigate the effects of PACs on altered gut microbiota under certain pathological conditions, a few studies using normal animals have been conducted. It was reported that dietary PACs for 6 days resulted in an ecological shift in the microbiome, dramatically increasing Lachnospiraceae, unclassified Clostridales, Lactobacillus, and Ruminococcus operational taxonomic units (OTUs) in crossbred female pigs. Further, intact parent PACs (dimer-pentamer) and major phenolic metabolites (4-hydroxyphenylvaleric acid and 3-hydroxybenzoic acid) were found in the feces [70]. It was reported that Lachnospiraceae and Ruminococcus are the major butyrate and propionate producers in human fecal samples [71], and butyrate can modulate oxidative stress in the colonic mucosa of healthy humans [72]. In a review article, butyrate was reported to lead to more specific and efficacious therapeutic strategies for the prevention and treatment of different diseases ranging from genetic/metabolic conditions to neurological degenerative disorders [73]. In particular, in a human study, transfer of intestinal microbiota from lean donors increased insulin sensitivity in individuals with metabolic syndrome along with levels of butyrate-producing intestinal microbiota, suggesting that intestinal microbiota should be developed as therapeutic agents to increase insulin sensitivity in humans [74]. If PACs have an ability to increase butyrate-producers, they may not only work for colonic health but also systemic health. Another study using weaned piglets revealed that dietary grape seed PACs improved the microbial diversity in ileal and colonic digesta, with the most abundant OTUs belonging to two phyla Firmicutes and Bacteroidetes [75]. The PACs also decreased abundance of Lactobacillaceae and increased that of Clostridiaceae, accompanied with improved intestinal mucosal barrier function and increased concentration of propionic and butyric acids in intestinal digesta. In a rat study where an 8-day oral gavage of grape seed PACs [monomeric (21.3%), dimeric (17.4%), trimeric (16.3%), tetrameric (13.3%) and oligomeric (31.7%)] was administered to normal female rats, the ratio of Firmicutes to Bacteroidetes at phylum level was lowered with increased plasma glucagon-like-peptide-1 level [76]. More recently, it was reviewed that PACs have a prebiotic and antimicrobial role favoring homeostasis of the intestinal environment, thus reducing the survival of Gram-negative bacteria that produce lipopolysaccharide (LPS) [77]. As LPS triggers the activation of the Toll-like receptor-4 (TLR-4) inflammatory pathway, PACs can minimize endotoxemia.

As for animal studies under pathological conditions, most studies applied high-fat diet (HFD) or high fat/high sucrose diet (HFHSD)-induced metabolic syndrome model animals. PAC-rich grape seed/pomace extract [78-80], PAC-rich cranberry extract [81], and apple PACs [82] showed improved symptoms of metabolic syndrome concomitant with altered gut microbial environment. Some studies revealed that PACs increase Akkermansia muciniphila [78, 80] or Akkermansia at the genus level [82], the former of which is a well-known beneficial gut bacterium that improves host metabolic functions and immune responses [83-89]. Accounting for 3-5% of the microbial community in healthy individuals, A. muciniphila is a mucinolytic bacterium found in the mucus layer of the human gut [90], and has the potential to restore mucus thickness and intestinal barrier integrity [91, 92]. The bacterium also has the ability to decrease the progression of many diseases such as obesity and type 2 diabetes mellitus [93, 94]. As such, A. muciniphila is considered a promising probiotic candidate [88]. At the phylum level, PACs could de-crease the ratio of Firmicutes/Bacteroidetes [79, 82]. The dominant gut microbial phyla are Firmicutes, Bacteroidetes, Actinobacteria, Proteobacteria, Fusobacteria, and Verrucomicrobia, with the first two phyla being the most common in healthy human individuals [95]. Phylum-level analyses of Firmicutes and Bacteroidetes have shown they are associated with obesity and that an increased population of Bacteroidetes, as well as a reduced population of Firmicutes, could improve obesity [96-100]; likely via de-pression of an increased capacity for energy harvest from the diet [99]. In a human study, the relative proportion of Bacteroidetes was decreased in obese people by comparison with lean people, and this proportion increased with weight loss on two types of low-calorie diets [101]. These findings indicate that obesity is associated with a microbial component, paving way for investigations into potential therapeutic implications of gut microbiota. Besides HFD- or HFHSD-fed animals, PACs normalized the imbalanced Firmicutes/Bacteroidetes ratio observed in OVX mice in a menopause model and prevented OVX animals from increased weight [102].  

If microbial degradation of PACs is hampered because of the low compound up-take by bacteria due to their huge molecular size, how exactly they affect gut microbiota becomes the primary concern. Some PACs exert antimicrobial activities by preventing bacterial adhesion to human cells [103, 104], with PAC-rich cranberry being used clinically as adjuvant therapy in preventing urinary tract infections [105]. It has also been reported that anti-adhesion activity could be challenging in the development of new antimicrobials that are able to withstand the increasing repertoire of bacterial resistance [106]. In dentistry, PACs are known to have specific antibacterial characteristics to attack periodontopathogenic bacteria (Porphyromonas gingivalis) but not oral commensal bacteria (Streptococcus salivarius) [107, 108]. PACs’ antibacterial activity in the oral cavity may be attributed to biofilm disrupting properties through interfering with a N-acylhomoserine lactone-mediated quorum sensing of the bacteria, which tightly regulates the expression of multiple virulence factors in opportunistic pathogenic Gram-negative bacteria [109, 110]. Thus, PACs could be one of the substances affecting gut microbiota via antibacterial activity. Further studies are needed to clarify the effect of PACs on the gut microbiota.

The aforementioned metabolic fate of PACs through the digestive tract and their health benefit effects associated with gut microbiota are summarized in Figs.2 and 3, respectively.

Reviewer 2 Report

I think the paper makes a lot of sense and yet I have to ask about the word intradigestive. According to Google it is an adjective meaning "between digestions of successive meals".  And Google gave me nothing when I asked the meaning of "intradigestive fate".  I am tempted to ask them to drop this word and instead say "metabloic fate". But maybe the reason the authors used the term intradigestive fate was to distinguish that it wasn't just the metabolites of PAC that matter (actually the commensals activity mattered a lot). So, I just ask that this terminology be checked to make certain readers know what it means. AND THAT READERS WILL NOT BE MISLED THAT THIS IS A PAPER ABOUT THE THINGS THAT HAPPEN BETWEEN FEEDINGS. 

Author Response

I think the paper makes a lot of sense and yet I have to ask about the word intradigestive. According to Google it is an adjective meaning "between digestions of successive meals".  And Google gave me nothing when I asked the meaning of "intradigestive fate".  I am tempted to ask them to drop this word and instead say "metabolic fate". But maybe the reason the authors used the term intradigestive fate was to distinguish that it wasn't just the metabolites of PAC that matter (actually the commensals activity mattered a lot). So, I just ask that this terminology be checked to make certain readers know what it means. AND THAT READERS WILL NOT BE MISLED THAT THIS IS A PAPER ABOUT THE THINGS THAT HAPPEN BETWEEN FEEDINGS.

Response to the comment

We appreciate your encouraging comment and helpful suggestion on the terminology. We totally agree with you and rephrased the title as follows; Metabolic fate of orally ingested proanthocyanidins through the digestive tract.

Reviewer 3 Report

I find the manuscript entitled "Intradigestive fate of orally ingested proanthocyanidins" interesting and a good contribution towards understanding the bioaccesibility and bioavailability of PACs. I think the manuscript is weel-written and clear and I recommend it for publication with just minor grammar/ spell check corrections.

Author Response

I find the manuscript entitled "Intradigestive fate of orally ingested proanthocyanidins" interesting and a good contribution towards understanding the bioaccesibility and bioavailability of PACs. I think the manuscript is weel-written and clear and I recommend it for publication with just minor grammar/ spell check corrections.

Response to the comment

We appreciate your encouraging comment very much. As for English grammar and spelling, we had our manuscript checked by a professional English editing agency.